



# An observation-constrained estimation of brown carbon aerosol direct radiative effects

Yueyue Cheng[1#], Chao Liu[1#], Jiandong Wang[1,*], Jiaping Wang[2], Dafeng Ge[2], Caijun Zhu[2], Jinbo Wang[2], and Aijun Ding[2]

[1] Collaborative Innovation Center on Forecast and Evaluation of Meteorological Disasters/China Meteorological Administration Aerosol-Cloud-Precipitation Key Laboratory, School of Atmospheric Physics, Nanjing University of Information Science and Technology, Nanjing 210044, China
[2] Joint International Research Laboratory of Atmospheric and Earth System Sciences, School of Atmospheric Sciences, Nanjing University, Nanjing, 210023, China

* *Correspondence to:* Jiandong Wang (jiandong.wang@nuist.edu.cn)

[#] These authors contributed equally: Yueyue Cheng, Chao Liu

**Abstract.** Brown carbon (BrC) is an organic carbon component with noticeable absorption in the ultraviolet and short visible wavelengths, which influences the global radiative balance. However, the assessment of BrC radiative effects remains a challenging task because of the scarcity of direct BrC observations and uncertainties in their chemical and optical properties. This study proposes a convenient method for estimating BrC radiative effects based on concise observational data. The light-absorbing properties of BrC obtained from aethalometer measurements and an optical separation method were combined with the simulated BrC optical properties to determine their mass concentrations. The aerosol optical depth (*AOD*) and mass concentration of PM$_{10}$ were used to constrain the total and other aerosol contents, and the optical properties and concentrations were estimated using an optical closure study. Such a state-of-the-art combination of measurements and numerical models provides the primary variables for radiative transfer simulations to estimate the BrC radiative effects. We use observations over four months (from July 1 to November 18, 2021) in Nanjing (a megacity in East China) as an example. During the observational period, BrC absorption constitutes 8.7–34.1% of the total aerosol absorption at 370 nm. In the atmosphere, BrC plays a warming role with its average instantaneous radiative forcing (RF) and standard deviation of 6.4 ± 3.4 W m$^{-2}$, 29.2% that of black carbon (BC). At the surface, the BrC-induced actinic flux (AF) attenuation was comparable to that caused by BC, accounting for over 40% of BC effects in the UV range and almost 20% in the visible range. Furthermore, the photosynthetically active radiation (PAR) caused by BrC is about 34.7 ± 9.7% that caused by BC. These findings provide valuable insights into the understanding of BrC radiative effects and indicate the importance and necessity of better observation and modeling of BrC properties.

## 1 Introduction

As a major component of atmospheric aerosol, carbonaceous aerosol can directly and indirectly influence the Earth's radiative budget (Bond et al., 2006; Laskin et al., 2015). Carbonaceous aerosol include organic carbon (OC) and black carbon (BC). Because of its strong light absorption in the visible and near-infrared regions, BC plays a crucial role as a climate warming agent (Moosmüller et al., 2011; Bond et al., 2013). Conversely, although OC is primarily thought to scatter aerosol and cool the atmosphere, recent studies have indicated that a collection of OC compounds also possess non-negligible light absorption (Alexander et al., 2008; Laskin et al., 2015). This subset of OC, also known as brown carbon (BrC), absorbs ultraviolet (UV)





and shortwave visible light, with its absorption diminishing as the wavelength increases (Chen and Bond, 2010; Kirchstetter and Thatcher, 2012), so BrC exhibits a significantly higher absorption Ångström exponent (AAE) than BC (Lack and Langridge, 2013; Mohr et al., 2013). Furthermore, previous studies have shown that BrC can contribute up to 10% to carbonaceous aerosol absorption at the visible wavelengths and up to 20–50% at shorter wavelengths (Chen et al., 2020; Zhang

et al., 2020; Pani et al., 2021). This substantial contribution leads to notable direct radiative effects (DREs) at the top of the atmosphere (TOA), with a global average value ranging from 0.1 to 0.6 $W·m^{-2}$ (Feng et al., 2013; Huang et al., 2018; Jo et al., 2016; Kirillova et al., 2014). These DREs offset the cooling effects caused by scattering-dominant aerosol to varying extents (Liu et al., 2014; Zhang et al., 2017b, 2021b). Lin et al. (2014) reported that the BrC DREs may be 27% to 70% of those attributed to BC. BrC is a carbonaceous component that has attracted significant interest recently; however, our understanding

of BrC and its effects are relatively limited and have great uncertainties.

Currently, materials such as humic-like substances, polycyclic aromatic hydrocarbons, and lignin are all considered BrC. The diversity in the chemical nature and microphysical properties of the BrC components introduces large variations in their content and optical properties (Andreae and Gelencser, 2006). Even in well-controlled chamber experiments, BrC light absorption is

influenced by a variety of factors (Chen and Bond, 2010; Cai et al., 2020; Zhong and Jang, 2014; Liu et al., 2016), such as combustion conditions (e.g., burning temperature), atmospheric acidity, $NO_x$ concentrations, and relative humidity. Liu et al. (2019) measured the optical properties of atmospheric water-soluble BrC in eastern China and reported BrC absorption coefficients of 9.4 $Mm^{-1}$ at a wavelength of 365 nm, much smaller than those observed in Beijing (14.0 $Mm^{-1}$) or Xi'an (19.6 $Mm^{-1}$) (Yan et al., 2015; Li et al., 2020). Choudhary et al. (2021) investigated BrC from paddy and wheat residue burning in

the source region of the Indo-Gangetic Plain and gave absorption coefficients at 405 nm of 134.8 $Mm^{-1}$ and 47.1 $Mm^{-1}$, respectively. Paraskevopoulou et al. (2023) indicated that the BrC absorption coefficients under intense residential wood burning conditions in Southeastern Europe can vary by a factor of 1.6 due to solvent differences. Thus, there are still significant uncertainties regarding the BrC chemicals and their optical properties.

Considerable attention has been devoted to estimating the global radiative effects of BrC by using numerical models (Feng et al., 2013; Wang et al., 2014, 2018b). These simulations mostly assume that a specific fraction of organic aerosol is BrC and use the measured minimum or maximum BrC absorption properties to perform radiative parameterization (Feng et al., 2013; Lin et al., 2014). For instance, Feng et al. (2013) assumed that 66% of primary organic aerosol from biofuels or biomass were BrC and absorptive, whereas Lin et al. (2014) assumed that all primary organic aerosol from biofuel and biomass sources, as

well as secondary organic aerosol produced by biogenic and anthropogenic emissions, were BrC. Chemical transport and radiative transfer models have been used to simulate the global or regional distributions of BrC and to quantify its radiative effects, both of which show significant differences, primarily because of limited constraints on BrC properties (Jo et al., 2016; Wang et al., 2014; Yan et al., 2018). However, direct large-scale and long-term observations of BrC are scarce. Zeng et al. (2020) addressed this issue by estimating the BrC DRE based on direct BrC observations collected from aircraft flights near



pole-to-pole latitudes. They discovered that BrC absorption contributes 7 to 48% of the TOA clear-sky instantaneous forcing by all absorbing carbonaceous aerosol. Although this methodology has great potential, direct atmospheric BrC observations are insufficient (Saleh, 2020).

      In summary, current BrC DRE estimations are based on either numerical modeling predictions based on specific assumptions
or well-planned and equipped field campaign measurements. Given the scarcity and challenges associated with *in situ* BrC observations, this study aims to propose a simplified but observation-constrained model to quantify BrC radiative effects; radiative forcing (RF) and influences on the photosynthetically active radiation (PAR) and actinic flux (AF) will all be considered and discussed. To achieve this, our estimation intends to combine the most readily available atmospheric and aerosol observations and fundamental numerical optical simulations and take observations over four months in Nanjing as an
example to evaluate BrC radiative effects relative to those of BC.

## 2 Sampling sites and observations

This study provides a general observation-constrained method for estimating BrC radiative effects by utilizing conventional observations that are easily accessible. First, aerosol properties from the ground-based measurements are obtained, including light absorption coefficients ($Abs$, in a unit of Mm$^{-1}$) from a multi-wavelength aethalometer, light scattering coefficients ($Sca$,
in a unit of Mm$^{-1}$) from a nephelometer, retrieved aerosol optical depth ($AOD$) from a Sun-photometer and the PM$_{10}$ mass concentration ($M_{PM10}$, in a unit of μg m$^{-3}$) from an aerosol particle size distribution sampler. A multi-wavelength aethalometer (Model AE-33, Magee Scientific Corporation Berkeley) continuously measured the aerosol $Abs$ at wavelengths of 370, 470, 520, 590, 660, 880, and 950 nm. The light absorption coefficients were obtained from the measured attenuation coefficients $Abs_{ATN}$ by correcting the effects due to filter loading and multiple scattering using the Eq. (1) (Schmid et al., 2006):

$$Abs_\lambda = \frac{Abs_{ATN}}{(C_{ref}+C_{scat}) \cdot R},$$  (1)

where R is the function for the filter-loading correction, and $C_{ref}$ is the multiple scattering correction factor (set to 4.26), following Coen et al. (2010). $C_{scat}$ represents the aerosol scattering correction, which is calculated by using the $Sca$ and scattering Ångström exponents at three wavelengths (450, 525, and 635 nm) obtained from an integrating nephelometer (Aurora™ 3000, Ecotech), following Arnott et al. (2005). The measured $Sca$ values were adjusted for hygroscopic growth
when $RH > 50\%$ (Zhang et al., 2015). A Sun/sky radiometer (CE318-T Photometer, CIMEL) was employed to obtain the retrieved $AOD$ from ultraviolet to near-infrared wavelengths centered at 440, 500, 670, 870, and 940 nm.

Nanjing, a megacity located in the Yangtze River Delta (YRD) region of China, is characterized by rapid growth and a dense population (Ding et al., 2013). Recent observational studies revealed the non-negligible light absorption of BrC in Nanjing
and its remarkable influence on climate change. The aethalometer, nephelometer, and Sun/sky radiometer observations were





performed at the SORPES station (118°57'10'' E and 32°07'14'' N), which is located in the Xianlin campus of Nanjing University in the western suburb region and is heavily impacted by human activity (Ding et al., 2013, 2016; Wang et al., 2018a). The $M_{PM10}$ were obtained from the China National Urban Air Quality Real-time Publishing Platform (http://113.108.142.147:20035/emcpublish/), and hourly results were adopted at the closest national monitoring station to the

SORPES (Xianlin University Town, 118°54'45" E and 32°06'10" N). The sampling period was from 1 June 2021 to November 18, 2021 (LT).

## 3. Extraction of BrC and non-BrC aerosol properties

Figure 1 illustrates the temporal variations in major parameters during the observational period, including $M_{PM10}$, $Abs$, $Sca$, and $AOD$. These observations were employed to determine the BrC concentrations and optical properties. As depicted in Fig.

1, these statistical observations describe different aerosol properties. Note that $AOD$ characterizes the accumulated aerosol extinction of the column and the other variables represent the aerosol properties at the surface. The hourly mean $M_{PM10}$ are displayed in Fig. 1(a), with daily mean values varying from 10 to 80 µg m$^{-3}$. The $Abs$ and $Sca$ reveal the ability of aerosol to absorb and scatter light, with daily mean values varying from 38 to 228 Mm$^{-1}$ at 525 nm and from 4 to 52 Mm$^{-1}$ at 520 nm, respectively. The $AOD$ describe the aerosol attenuation effects on light, with daily mean values varying from 0.1 to 1.2 at 520

nm.

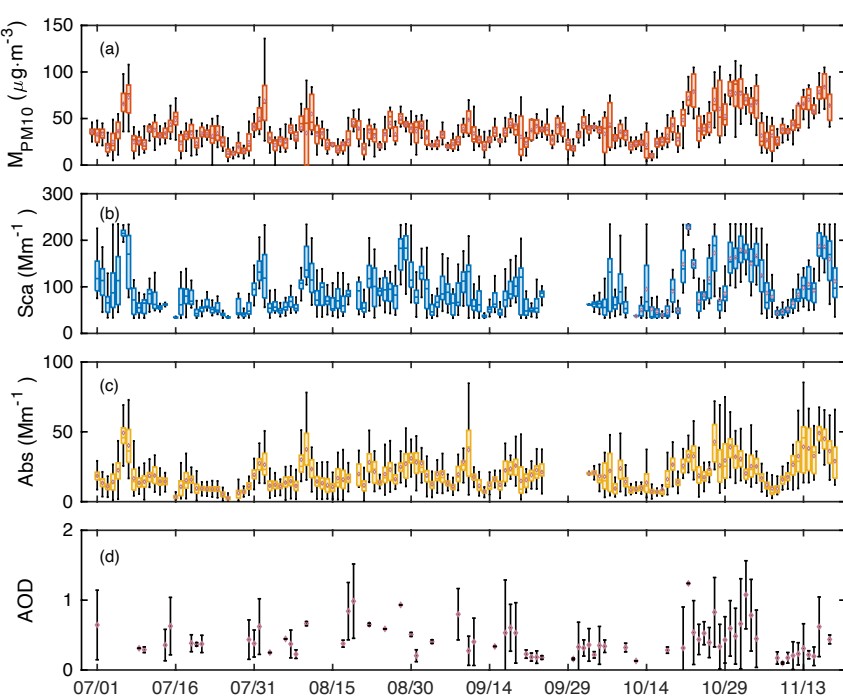



**Figure 1. Time series of (a) $PM_{10}$, (b) $Sca$ at 525 nm, (c) $Abs$ and (d) $AOD$ at 520 nm during the observations. The boxes in (a), (b), and (c) indicate the 25% and 75% quantile of daily observation data, and the tentacles whiskers indicate 5% and 95%. The line near the middle of each box indicates the median of the sample, and the red diamond sign symbol in the box indicates the day's mean value of the data within a day.**

The above observations were used to constrain the background aerosol properties and distinguish BrC and BC components. To evaluate the radiative effects of BrC in the atmosphere, a combination of observations and well-accepted microphysical and optical characteristics were adopted (Fig. 2). First, aerosol properties, including $M_{PM_{10}}$, $Abs$, $Sca$, and $AOD$, were obtained from direct observations corresponding to the red part of Fig. 2. The mass absorption cross section (MAC), mass extinction cross section (MEC), single scattering albedo (SSA), and asymmetry factor (ASY) for each type of aerosol are from the modeled aerosol optical properties, which are illustrated in green (Fig. 2). Finally, the measured and simulated aerosol properties were combined using an optical closure study to partition the various aerosol properties, that is, $AOD$, ASY, and SSA. With the derived aerosol properties, the radiative transfer model (Library for Radiative Transfer Model, LibRadTran Model) can be used to determine the BrC radiative effects and their contribution to the relative carbonaceous aerosol effects.

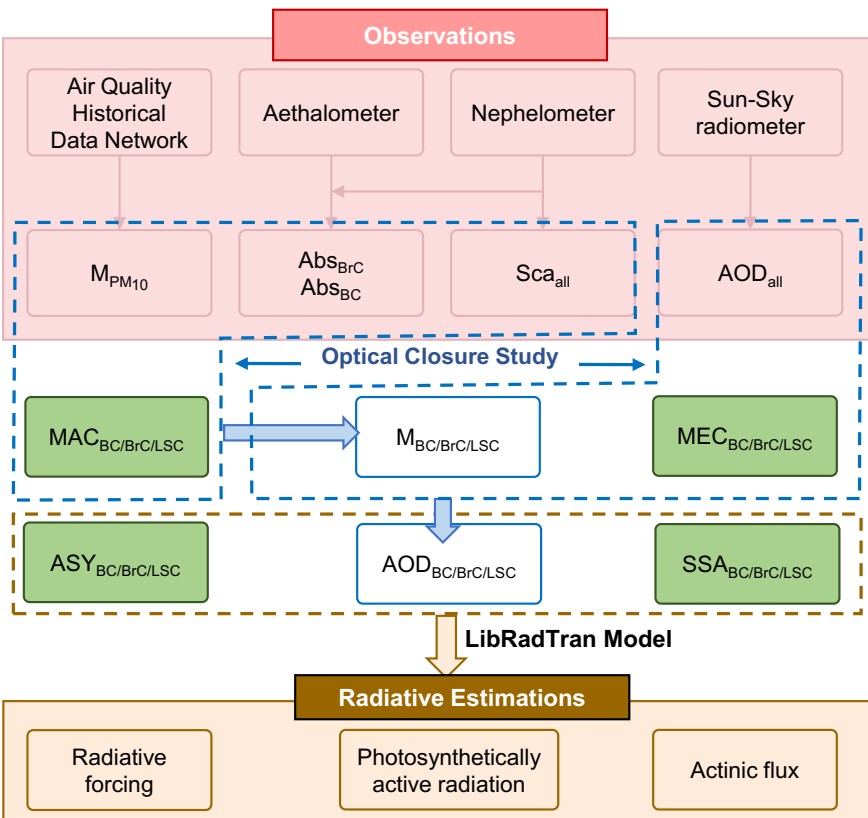

**Figure 2. Flowchart of BrC radiative effect estimation.**






## 3.1 Extraction of BrC absorption

The optical properties of BrC, particularly its absorption, are pivotal factors for estimating radiative forcing (Lu et al., 2015; Saleh et al., 2015). Initially, we extracted the absorptions of BC and BrC from the AE-33 observations following the absorption Ångström exponent ($AAE$) segregation method (Lack and Langridge, 2013; Mohr et al., 2013; Shen et al., 2017), which
considers the distinct absorption spectral dependence of BC and BrC. Simplistically, the AAE derived from the AE-33 measurements can be conceptualized as the average of the BC and BrC absorption in the form of

$$AAE = \frac{-ln(\frac{Abs_{\lambda 1}}{Abs_{\lambda 2}})}{ln(\frac{\lambda 1}{\lambda 2})} . \tag{2}$$

By assuming that the absorption at longer wavelengths (specifically, 880 nm considered here) is solely due to BC (Virkkula et al., 2015) and a constant BC AAE of 1 (i.e., $AAE_{BC} = 1$) (Lack and Cappa, 2010; Liakakou et al., 2020; Kaskaoutis et al., 2021;
Zhang et al., 2021a), BC absorption at shorter wavelengths ($Abs_{\lambda\_BC}$) can be derived at 880 nm ($Abs_{880\_BC}$, which equals to $Abs_{880}$ directly from AE-33) as follows:

$$Abs_{\lambda\_BC} = Abs_{880} \times \left(\frac{880}{\lambda}\right)^{AAE_{BC}} . \tag{3}$$

Then, the BrC absorption coefficients ($Abs_{\lambda\_BrC}$) are obtained as the result of the absorption coefficients measured by the aethalometer ($Abs_{\lambda}$) minus the light absorption coefficients of BC ($Abs_{\lambda\_BC}$) at a reference wavelength ($\lambda$):
$$Abs_{\lambda\_BrC} = Abs_{\lambda} - Abs_{\lambda\_BC} . \tag{4}$$

Following this segregation, Fig. 3 shows the time series of the daily average absorption coefficients of BrC and BC at 370 nm ($Abs_{370\_BrC}$ and $Abs_{370\_BC}$) during the observation period, along with the fraction of BrC absorption to total aerosol absorption ($P_{370}$) at the same wavelength. The daily mean $Abs_{370\_BrC}$ varies from 1.1 to 13.0 Mm$^{-1}$ with a mean value of 4.4 Mm$^{-1}$, which falls within the same order of magnitude but is lower than those reported in Beijing (Du et al., 2014; Yan et al., 2015;
Cheng et al., 2016) and Xi'an (Yuan et al., 2016a; Zhu et al., 2021). BrC absorption exhibit a variation similar to that of BC at shorter wavelengths. $Abs_{370\_BrC}$ shows slight seasonal variations with a higher average value in autumn (4.9 Mm$^{-1}$) than in summer (2.8 Mm$^{-1}$).



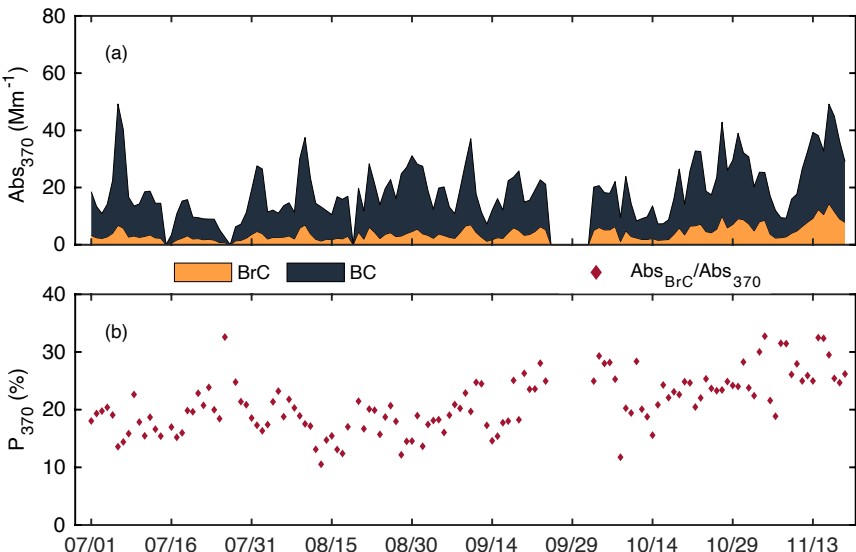

**Figure 3. Variations of daily average (a) $Abs_{370\_BC}$ and $Abs_{370\_BrC}$ and (b) $P_{370}$ during the measured period.**

Overall, BrC aerosol accounts for 8.7 to 34.1% of the total carbonaceous aerosol absorption at 370 nm, which is similar to the monthly mean fractions from eight sites reported by Wang et al. (2016). The mean $P_{370}$ in summer and autumn were 16 and 23%, respectively. The $P_{370}$ in summer is lower than that reported in Xi'an of 23.8%, but higher than in the PRD region of 165 11.7% (Yuan et al., 2016a; Shen et al., 2017). The $P_{370}$ tends to be higher in more polluted urban areas, such as Xianghe (China) and Mexico City, which has BrC absorption fractions of 30 to 40% at 400 nm (Barnard et al., 2008; Yang et al., 2009). In compared to the investigation of Wang et al. (2018) in Nanjing (mean $Abs_{370\_BrC}$ and $P_{370}$ of 5.9 Mm⁻¹ and 19.6 % in summer, respectively), our data show lower $Abs_{370\_BrC}$ and higher $P_{370\_BrC}$. Almost all $P_{370}$ can reach to 20 % in October and November, which cannot be ignored when estimating the subsequent radiative effects.

**3.2 Aerosol optical properties**

The aforementioned conventional *in situ* measurements provide some of the overall aerosol optical properties and contents, that is, mass concentration ($M$), $AOD$, $Abs$, and $Sca$, whereas those of each aerosol component are needed to investigate their corresponding contributions. We attempted to assign the known total aerosol properties to BrC using the above parameters, in conjunction with an optical closure study. First, aerosol light absorption was assumed to be solely attributed to absorptive 175 carbonaceous aerosol (BC and BrC) because the dust contribution was much less in Nanjing during this period (Feng et al., 2023). Second, in addition to BrC and BC, the remaining scattering-dominant aerosol (e.g., mainly sulfate and nitrate at our observation site) possess similar optical properties and will therefore be treated together as a non-absorbing type. Consequently, we adopted a three-component aerosol model (BrC, BC, and pure-scattering components) that was assumed to



be externally mixed (Yuan et al., 2016b). The absorption characteristics of BrC and BC were determined based on the

observations described in Section 3.1, and their mass concentrations were calculated using Mie numerical simulations.

In the Mie numerical simulations, the adopted BrC size distribution reported by Kirchstetter et al. (2004) was adopted, i.e., a lognormal size distribution with a mean radius of 80 nm and a standard deviation of 1.4. The density of BrC is set to be 1.2 g·cm⁻³ (Lin et al., 2014). The real part of the BrC refractive index (RI) is set to be 1.55, based on Chen and Bond (2010).

However, the reported BrC RI imaginary part ($k_{BrC}$) exhibits a sharp increase towards shorter wavelengths with significant differences from different studies (Andreae and Gelencser, 2006). Figure 4 illustrates the BrC RI at wavelengths between 300 and 800 nm, retrieved from either *in situ* or laboratory observations (Alexander et al., 2008; Chakrabarty et al., 2010; Chen and Bond, 2010; Kirchstetter and Thatcher, 2012; Hoffer et al., 2016; Liu et al., 2016; Shamjad et al., 2016; Zhang et al., 2017a). The $k_{BrC}$ values of Alexander et al. (2008) and Chakrabarty et al. (2010) were for individual amorphous carbon spheres

produced from the East Asia-Pacific outflow and chamber studies, respectively. Liu et al. (2016) obtained the $k_{BrC}$ during a biomass-burning event in Colorado, USA, and Shamjad et al. (2016) sampled atmospheric BrC from Kanpur, India. The $k_{BrC}$ in Kanpur (average $k_{BrC}$ of 0.037) was found to absorb fourfold more at 405 nm wavelength compared to the USA (average $k_{BrC}$ value of 0.009). Among the above $k_{BrC}$, the group reported by Shamjad et al. was considered to have moderate values. Overall, the uncertainty of the real part of the BrC RI was less than that of the imaginary parts, showing differences of over an

order of magnitude. Thus, selecting a suitable set of RI is crucial for representing the optical properties of BrC, which significantly influence our optical closure. To better restore the real situation of BrC in the atmosphere, we adopted the $k_{BrC}$, which was linearly extrapolated from the investigation by Shamjad et al. (2016) in an urban Indian city, Kanpur (i.e., 0.041i at 370 nm, 0.026i at 470 nm, 0.018i at 520 nm, 0.0105i at 590 nm, and 0.004i at 660 nm).

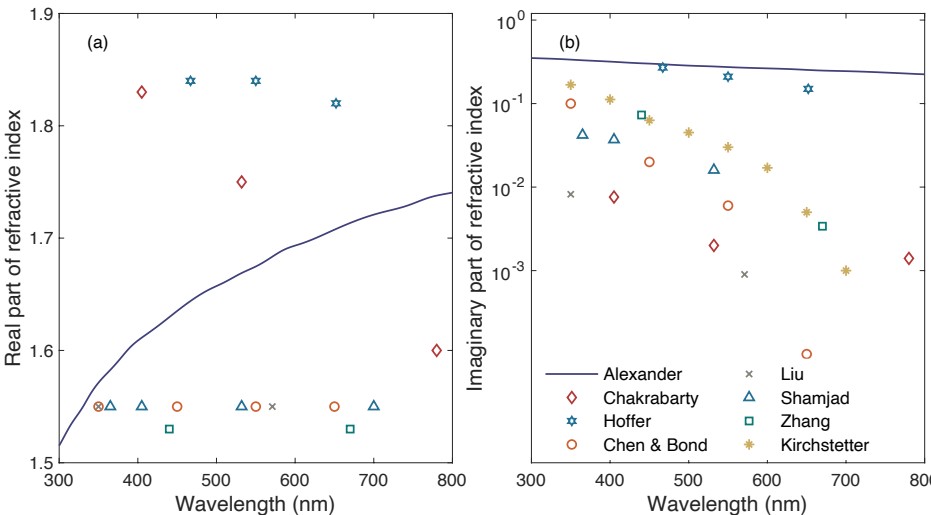






**Figure 4. The real (a) and imaginary part (b) of the BrC aerosol RIs. The sample case of Chen and Bond (2010) shown here is for methanol-soluble OAK_L_360 emitted from solid fuel pyrolysis. We derive the $k_{BrC}$ based on $\frac{\rho\lambda\sigma}{4\pi}$, where ρ is the density of BrC, 1.65 g·cm⁻³.**

Meanwhile, we utilized the BC and sulfate (scattering-dominant) size distributions (with mean radius of 11.8 and 69.5 nm and standard deviations of 2 and 2.03, respectively) from the Optical Properties of Aerosols and Clouds (OPAC) database (Hess et al., 1998). The density of BC and scattering aerosol are assumed to be 1 and 1.7 g·cm⁻³, respectively. The RI of BC and sulfate aerosol at 550 nm are reported as 1.80-0.54i and 1.50-10⁻⁷i (Cheng et al., 2006; Ma et al., 2011; Yuan et al., 2016b), respectively, with minimal variation of imaginary part of RI across wavelengths. Subsequently, Mie theory was applied to

obtain the average optical properties of BrC, BC, and the pure-scattering components (Bohren and Huffman, 1983), respectively, which include their MAC, MEC, SSA, and ASY under the assumption of homogeneous spherical particles.

Based on the segregated *Abs* of light absorptive carbon aerosol (LAC, i.e., BC and BrC) in Section 3.1 and the simulated optical properties in Section 3.2, the mass concentration of BrC ($M_{BrC}$) can be estimated as

$$M_{BrC} = \frac{Abs_{\lambda\_BrC}}{MAC_{\lambda\_BrC}}, \qquad (5)$$

where $\lambda$ is set to be 520 nm unless otherwise mentioned. $M_{BC}$ can be given values similar to the corresponding values for the BC. This study assumed that the mass concentration of $PM_{10}$ is the sum of all aerosol mass concentrations; therefore, the mass concentration of pure light-scattering aerosol ($M_{LSC}$) can be determined using Eq. (6).

$$M_{LSC} = M_{PM_{10}} - M_{BrC} - M_{BC} . \qquad (6)$$

Thus far, the *M* values of all aerosol species on the surface have been obtained. Daily *AOD* observations were used to determine the mass concentration distributions in the vertical direction. A uniformly distributed and constant aerosol concentration was assumed for all layers within the height of the aerosol boundary layer (*H*) (Yan et al., 2015). Given that $AOD_{All}$ is measured by a Sun-sky radiometer under a real atmospheric background, where the absorption by light-absorbing particles can be enhanced due to the uptake of water during the aging process, a factor of hygroscopicity (Kappa factor, *f*)

corresponding to atmospheric relative humidity is implemented to account for this enhancement (Kirillova et al., 2014). The *H* is given by Eq. (7).

$$H = \frac{AOD_{All}}{f \times (M_{BC} \times MEC_{BC} + M_{BrC} \times MEC_{BrC} + M_{LSC} \times MEC_{LSC})}, \qquad (7)$$

where $AOD_{All}$ and $MECs$ are from 520 nm. The *H* and *M* values of BC, BrC, and pure light scattering aerosol (LSC) are shown in Fig. 5. The *H* has mean values of 2.1 and 1.5 km during the summer and autumn, respectively. Higher aerosol boundary

heights during summer create favorable conditions for aerosol diffusion. There are slight seasonal fluctuations in $M_{BC}$, with average values of 1.2 and 1.5 µg·m⁻³ recorded during the summer and autumn, respectively. Conversely, $M_{BrC}$ exhibit a relatively higher difference between seasons, with average values of 2.0 and 3.2 µg·m⁻³ observed during the summer and autumn, respectively. Furthermore, $M_{LSC}$ exhibit a considerable seasonal variation as presented in Fig. 6b, with average





concentrations of 33.3 and 48.1 μg·m$^{-3}$ during the summer and autumn, respectively. The aforementioned seasonal variations
in aerosol concentrations can be partially attributed to the significant role of abundant summer rainfall in wet aerosol removal.

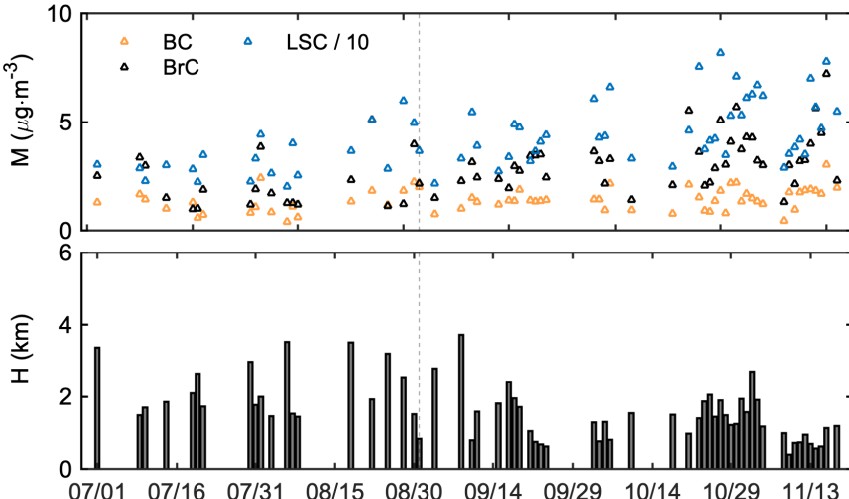

**Figure 5. Aerosol optical closure results of the observation period. Where, (a) illustrates the mass concentrations of BrC, BC, and LSC; (b) illustrates the height of aerosol diffusion boundary and the dotted grey line represents the division between summer and**
**autumn.**

Finally, the optical properties of each aerosol species (BC, BrC, and LSC) were determined for further radiative estimations.
Taking BrC as an example, the $AOD$ ($AOD_{BrC}$) can be calculated using Eq. (8) and its SSA and ASY were adopted from the
Mie simulation:

$$AOD_{BrC} = H \times M_{BrC} \times MEC_{BrC}. \tag{8}$$

BC and LSC were assigned similarly to the corresponding values of $AOD$, SSA, and ASY. To validate the mass concentration
segregation result, the calculated total aerosol scattering coefficients ($Sca_{cal}$) were obtained using Eq. (9), and then compared
with the observed $Sca$ ($Sca_{obs}$) to examine the relative errors in the concentration closure experiment. The $Sca$, $Abs$, and $MSC$
in Eq. (9) was 525 nm:

$$Sca_{cal} = M_{BC} \times MSC_{BC} + M_{BrC} \times MSC_{BrC} + M_{LSC} \times MSC_{LSC}. \tag{9}$$

The consistency between the $Sca_{cal}$ and $Sca_{obs}$ results indicate the feasibility of the optical closure study results (Fig. 6).





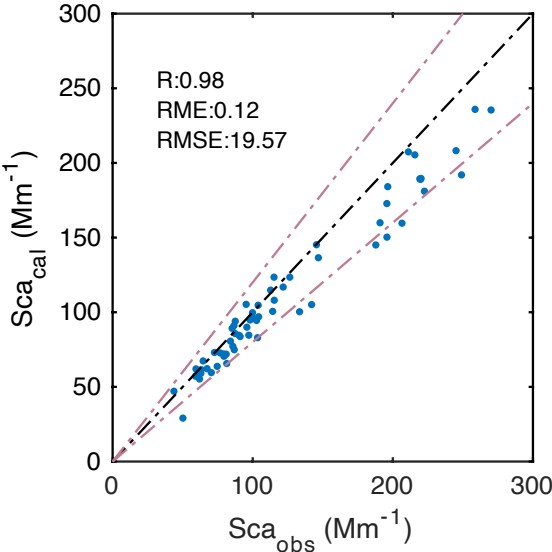

**Figure 6.** A comparison of observed $Sca$ ($Sca_{obs}$) and calculated $Sca$ ($Sca_{cal}$), the red line is the 20% relative error line.

## 4 BrC radiative effects

Based on the BrC, BC, and remaining aerosol properties from a coupling of observations and modeling, the LibRadTran was then used to calculate the BrC radiative effects (Mayer and Kylling, 2005; Ma et al., 2019). LibRadTran was performed three times for aerosol scenarios with all aerosol, without BrC, and without BC, and their differences were used to determine the contribution of BrC or BC. Specifically, we used the one-dimensional discrete ordinate radiative transfer (DISORT) model as the radiative transfer solver, and the solar spectrum ranging between 300 and 2500 nm was divided into seven spectral bands

centered at 370, 470, 520, 590, 660, and 950 nm, following Lu et al. (2020). The average aerosol optical properties, that is $AOD$, SSA, and ASY, of each band were obtained from the optical closure study mentioned above and were used to estimate the radiative effects. The spectral surface albedo followed Liao et al. (1999) and a solar zenith angle of 60º is used. To examine the aerosol radiative effects, we need to specify a vertical aerosol distribution to clarify the aerosol profiles (Yu et al., 2017; Lu et al., 2020). All aerosol were assumed to be uniformly distributed vertically within the aerosol diffusion height $H$.


Figure 7 shows the daily average BrC and BC RF estimations at the bottom of the atmosphere (BOA), at TOA, and in the atmosphere (AT). The RF within wavelengths of 300–700 nm and 300–2500 nm are illustrated in the left and right panels, respectively. In the wavelength range of 300–700 nm, the daily mean BC RF at the BOA varies from -3.6 to -33.6 W m⁻² with mean values of -15.8 ± 7.7 W m⁻² and daily BrC RF varies from -1.4 to -14.7 W m⁻², with mean values -5.4 ± 2.9 W m⁻². In

the AT, the daily mean BC and BrC RF vary from 4.0 to 38.4 and from 0.8 to 8.6 W m⁻², with mean values of 17.7 ± 8.8 and 3.1 ± 1.7 W m⁻², respectively. The BC and BrC RFs at the TOA are estimated to be 1.9 ± 1.1 and -2.3 ± 1.2 W m⁻², respectively. In the wavelength range of 300–2500 nm, the daily mean BC and BrC RF at the BOA range from -5.2 to -48.4 and from -1.7



to -17.5 W m$^{-2}$, respectively, with mean values of -22.7 ± 11.1 and -6.4 ± 3.5 W m$^{-2}$, and those in the AT range from 5.8 to

55.5 and from 1.0 to 10.7 W m$^{-2}$, respectively, with mean values of 25.7 ± 12.8 and 3.8 ± 2.1 W m$^{-2}$. The BC and BrC RFs at

the TOA are estimated to be 3.0 ± 1.7 and -2.6 ± 1.4 W m$^{-2}$, respectively. Accordingly, the fractional BrC RF relative to BC

RF (RRF) in the wavelength range of 300–2500 nm at the BOA is 29.2 ± 8.2%, and in the AT is 15.2 ± 4.3%. The RRF at the

BOA is 35.0 ± 9.8% and in the AT is 17.8 ± 5.0% when considering the UV–VIS range (300–700 nm). These values fall within

the range of the reported values in China, with average values ranging from 15% to 26%  (Huang et al., 2018; Li et al., 2020).

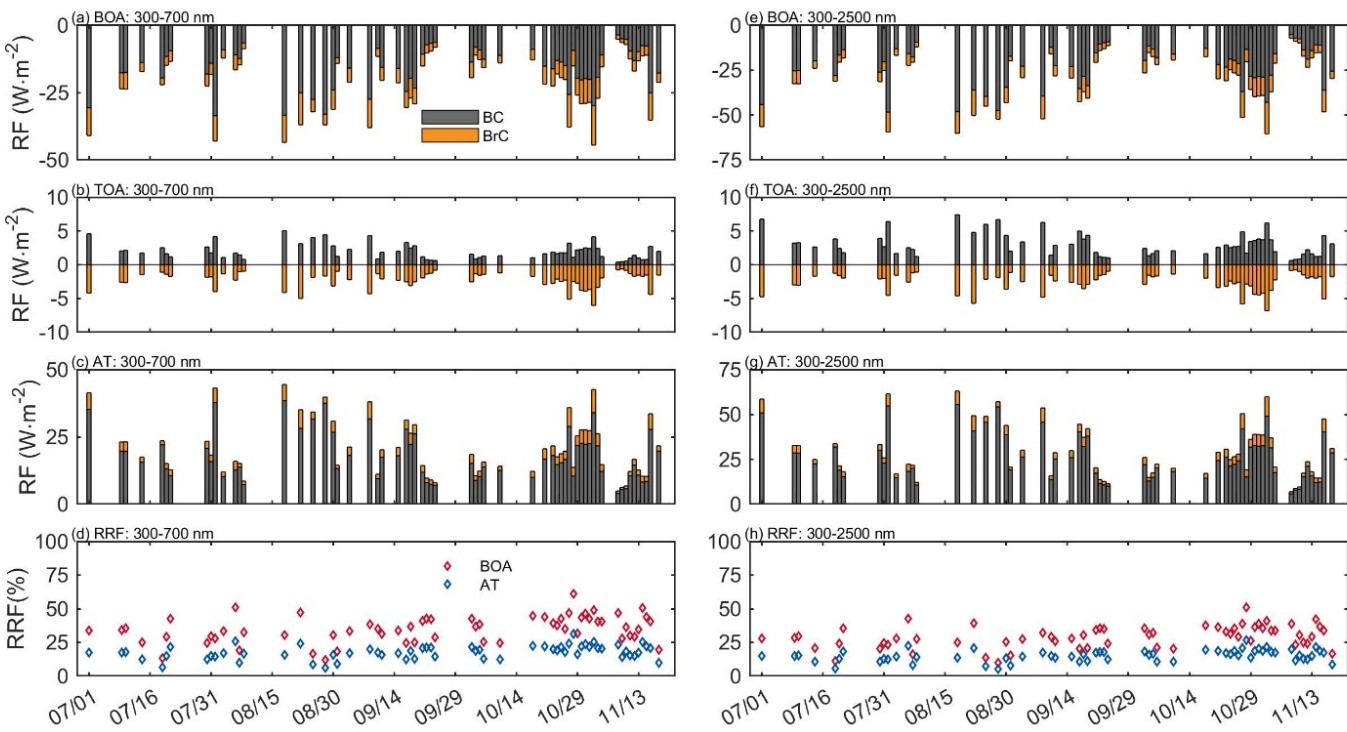


**Figure 7. The daily BrC and BC RFs at the BOA, at the TOA, and in the AT integrated in the wavelength range of 300–700 nm are shown in the (a), (b), and (c) respectively, as well as the daily fractional BrC RF relative to BC RF (RRF) is depicted in the (d). Corresponding results integrated in the wavelength range of 300–2500 nm are shown in the (e)–(h).**

BrC has strong absorption in the UV and shorter spectral regions, which efficiently reduce the AF and thus affect atmospheric

photochemistry and tropospheric ozone production (Jacobson, 1998; Mohr et al., 2013). Figure 8a and 8b demonstrate that

both BC and BrC decreased the AF in the UV range (300–400 nm) and visible range (400–700 nm) range, which is consistent

with an investigation conducted in Guangzhou (Li et al., 2020). In the UV range, we estimate the BrC-induced and BC-induced

AF to be -2.1×10$^{15}$ ± 0.9×10$^{15}$ and -8.8×10$^{14}$ ± 4.6×10$^{14}$ photons s$^{-1}$ cm$^{-2}$, respectively, and in the visible range to be -

10.2×10$^{15}$ ± 4.5×10$^{15}$ and -1.9×10$^{15}$ ± 1.1×10$^{15}$ photons s$^{-1}$ cm$^{-2}$, respectively. In addition, the attenuation effects of BrC

relative to that of BC (RAF) are significant, at 43.1 ± 11.7% in the UV range and 18.5 ± 5.4% in the visible range (Fig. 8c).





These findings suggest that BrC has a substantial effect on atmospheric photochemistry. The impacts of BrC and BC on AF were investigated on November 10, 2021, with the considerable RAF in the UV range (300–400 nm) and visible range (400–700 nm) ranges. Figure 8d shows the BrC-induced and BC-induced AF in the vertical direction and both of them exhibit

decreasing trend from the surface upwards. The BrC-induced AF change from negative to positive at certain height. The BC-induced AF always keep positive from the surface upwards. The differences are relevant to their differences in absorptive capacity.

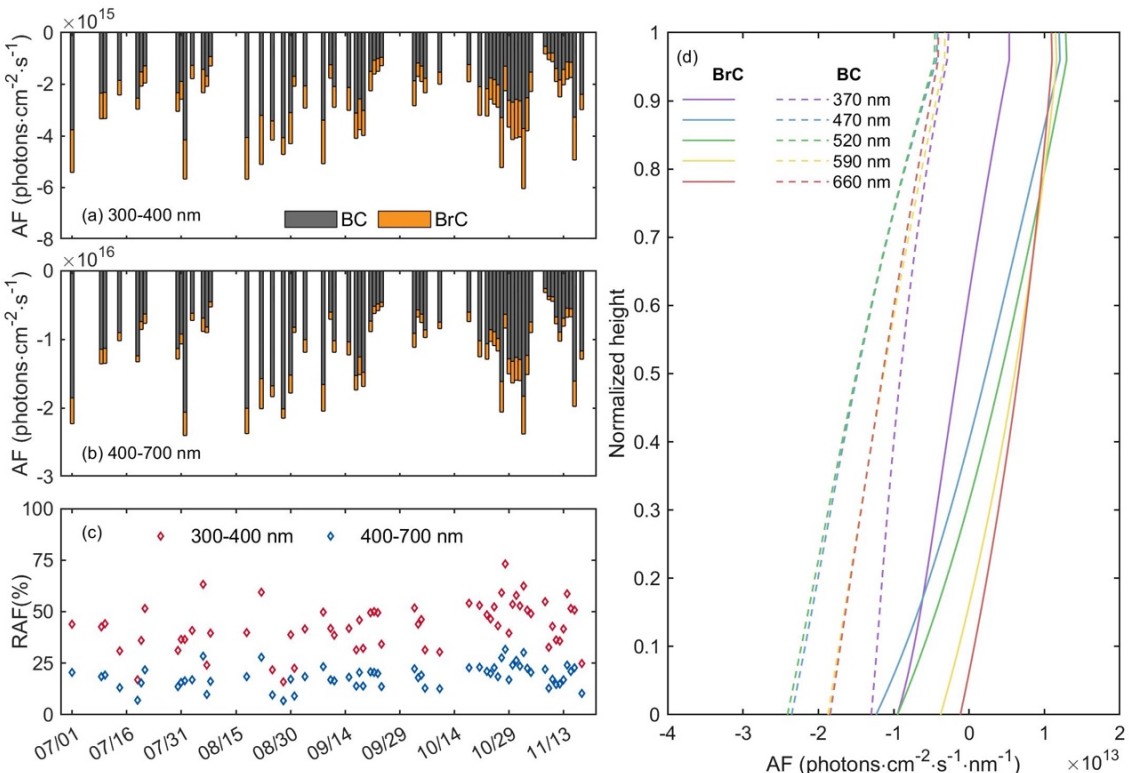

**Figure 8. The daily average AF is induced by BrC and BC (a) in the UV wavelength range (300–400 nm) and (b) in the visible wavelength range (400–700 nm). Accordingly, panel (c) illustrates the BrC-induced AF relative to BC-induced AF in the aforementioned two wavelength range. Panel (d) shows the vertical profile of BrC-induced and BC-induced AF at wavelengths of 370, 470, 520, 590, and 660 nm. Here the height is normalized.**

The impact of BrC-induced AF in the UV range (300–400 nm) is more significant than in the visible range (400–700 nm) range. Thus, BrC absorption can lead to a general reduction in $NO_2$ photolysis rates, resulting in the maximum decrease in surface $O_3$ concentrations (Jo et al., 2016). This is particularly relevant for areas with substantial biomass burning and biofuel emissions, where BrC can significantly affect air quality and climate. Another concern regarding BrC is its potential impact on the photosynthetically active radiation (PAR) used in the process of photosynthesis, which refers to the spectral range of

solar radiation of 400–700 nm. Figure 9 illustrates the PAR induced by both BrC and BC aerosol at the surface. During the





observation period, the direct attenuation of PAR induced by BrC and BC is estimated at -4.7 ± 2.5 and -13.9 ± 6.8 W m$^{-2}$, respectively, which slows down plant photosynthesis rates and reduces vegetation productivity. Figure 9b presents the proportion of PAR induced by BrC relative to that induced by BC (RPAR), with an average and standard deviation value of 34.7 ± 9.7%. These findings highlighted the significant impact of BrC on vegetation and its potential role in the environment.


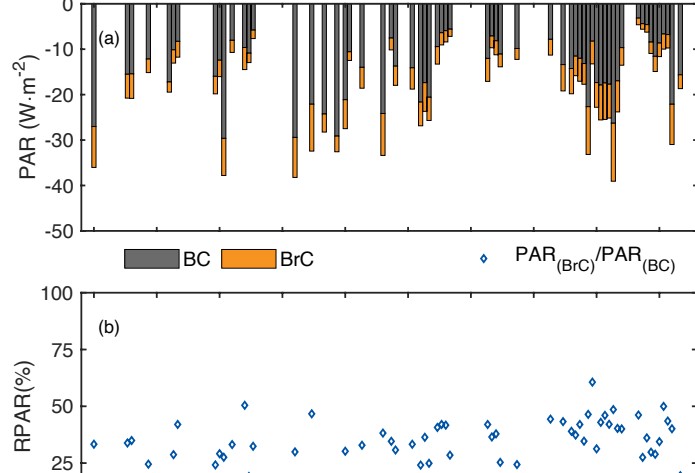

**Figure 9. The daily average (a) BrC-induced PAR and BC-induced PAR, as well as (b) fractional BrC-induced PAR relative to BC-induced PAR.**

Based on the above daily results for RRF, RAF, and RPAR, we investigated their corresponding monthly averages at the BOA (Fig. 10) and found that AF is the most contributed BrC-induced radiative effect among the aforementioned radiative effects. In addition, RRF, RAF and RPAR were all somewhat larger in autumn than in summer, which was related to the seasonal differences in BrC absorption and aerosol diffusion height.



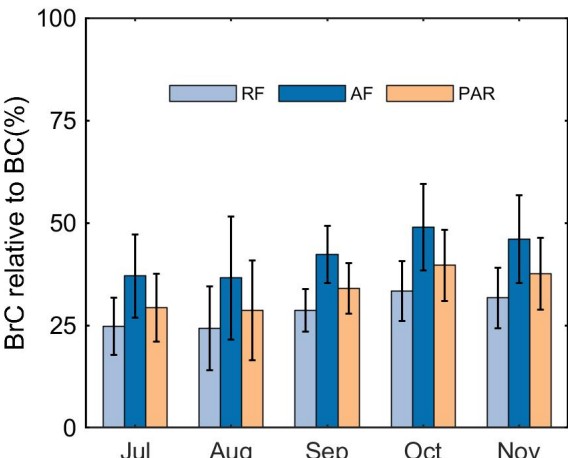

**Figure 10. BrC radiative effects relative to those of BC. The RF results are integrated in the wavelength range of 300–2500 nm and the AF results are integrated in the wavelength range of 300–400 nm.**

## 5 Conclusions and future work

In this study, we propose an observationally constrained and convenient approach for estimating the BrC radiative effects. A state-of-the-art combination of measurements and numerical models provides the primary variables for radiative transfer simulations to estimate the BrC radiative effects. The light-absorbing properties of BrC obtained from aethalometer measurements and an optical separation method were combined with the simulated BrC optical properties to determine their mass concentrations. Optical closure is performed to obtain the concentrations and optical information for BrC, BC and non-absorbing aerosol. The *AOD* and mass concentration of $PM_{10}$ were used to constrain the total and other aerosol contents. Then, radiative transfer simulations are employed to estimate the BrC radiative effects. Subsequently, observations over four months in Nanjing are used as an example to quantify the radiative effects of BrC. During the observational period, the absorption of BrC and BC at 370 nm is on the same order of magnitude, and their ratios range from 8.7% to 34.1%, aligning with values reported in previously studies. The BrC-induced instantaneous RF accounted for approximately $29.2 \pm 8.2\%$ that of BC at the surface, and $15.2 \pm 4.3\%$ in the atmosphere. The strong absorption of BrC in the near-ultraviolet and visible regions significantly weakened the PAR and AF at the surface, underscoring its non-negligible contribution to climate. These findings provide valuable insights into the understanding of BrC radiative effects and indicate the importance and necessity of better observation and modeling of BrC properties.

## Author contribution

Jiandong Wang and Chao Liu designed and directed the study. Yueyue Cheng contributed to data analysis and wrote the first draft of this paper. Jiaping Wang, Dafeng Ge, Caijun Zhu and Jinbo Wang collected data. Jiandong Wang, Jiaping Wang, Chao Liu and Aijun Ding contributed the data interpretation and review of the paper.



**Competing interests**

The authors declared that they have no conflict of interest.


**Acknowledgements**

This work was supported by National Key R&D Program of China (2022YFC3701000, Task 5, Jiandong Wang), and the National Natural Science Foundation of China, 42075098 (Jiandong Wang) and 42005082 (Jiaping Wang).

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
