# Peer review of "An observation-constrained estimation of brown carbon aerosol direct radiative effects"

_EGUsphere, 2023_

## Referee Comment (RC2)

The manuscript employs a hybrid method to estimate brown carbon (BrC) radiative effects using a combination of aethalometer measurements of aerosol absoprtion, an optical separation method, simulated BrC optical properties, and a radiative transfer model. As BrC is still poorly characterized in the field and largely ignored by many chemical transport models and climate models, the presented results contribute to our understanding of its radiation and climate significance. The manuscript fits the scope of ACP very well. I have some comments below for the authors to address.

1.  Line 143-145: the authors set the $AAE_{BC}=1$ to calculate the absorption contributions of BrC at different wavelengths. However, existing research suggests that the AAE of BC in the atmosphere varies within a certain range due to factors such as mixing state and morphology (Lack and Cappa, 2010; Zhang et al., 2020b). To comprehensively assess BrC's absorption contributions, different $AAE_{BC}$ values could be set, and some sensitivity analyses are needed to evaluate the impact of this parameter on the study results.

    *Lack, D. A. and Cappa, C. D.: Impact of brown and clear carbon on light absorption enhancement, single scatter albedo and absorption wavelength dependence of black carbon, Atmos. Chem. Phys., 10, 4207–4220, https://doi.org/10.5194/acp-10-4207-2010, 2010.*
    *Zhang, X., Mao, M., Yin, Y., and Tang, S.: The absorption Ångstrom exponent of black carbon with brown coatings: effects of aerosol microphysics and parameterization, Atmos. Chem. Phys., 20, 9701–9711, https://doi.org/10.5194/acp-20-9701-2020, 2020.*

2.  Line 178-179: In the optical closure calculation, a three-component aerosol model (BrC, BC, and pure-scattering components) was chosen with the assumption of an external mixing state. However, internal mixing between chemical components has been confirmed as a crucial factor influencing aerosol optical properties. It is recommended that the authors consider the internal mixing state to comprehensively evaluate the aerosol optical properties. Alternatively, the authors could explicitly state the reasons for choosing the external mixing model. This discussion will enhance the reliability of its results.

3.  Line 209-210: In this study, it is crucial to address whether the Mie numerical simulation is based on the spherical assumption, as this may introduce biases in optical calculations for non-spherical particles. Particularly, a spherical Mie model tends to significantly overestimate the light absorption of fractal BC particles. The numerical simulation of aerosol optical properties is intricately linked to the accuracy of mass distribution among different components. It is recommended that the potential biases introduced by the spherical assumption should be discussed.

4.  As shown in Figure 4, the uncertainty in the Refractive Index of BrC is notable. The authors should provide additional details explaining the reason for choosing the RI reported by Shamjad et al. (2016). It would be beneficial to illustrate the factors such as the similarity in organic aerosol composition and sources between the two cities. This additional information will enhance the understanding of the selection criteria for the RI values.

5.  Line 205-207: The data utilized in the study are relatively dated, and it is recommended to

consider relevant data from recent studies. The inclusion of more recent data would enhance the timeliness and relevance of the findings.

6. Line 207: The density of BC is commonly reported as 1.8 g·cm$^{-3}$ in the literature. It is advised to verify the accuracy of the statement indicating a density of 1.0 g·cm$^{-3}$ for BC in this context.

7. The calculation of radiative forcing (RF) in this study is unclear regarding whether it considers only direct radiative forcing or also includes indirect radiative forcing. Some clarification is needed.

8. English language needs to be further polised. Some necessary edits in the abstract:
   - L15, convenience -> efficient, concise -> available
   - L23, BrC induces a warming effect with an average instantaneous radiative forcing (RF) of 6.4 ± 3.4 W m-2, corresponding to 29.2% of the BC RF.
   - that of black carbon (BC).
   - L26, you may want to say "PAR attenuation".

---

## Author Comment (AC1)

**Response to Reviewer #1 (EGUSPHERE-2023-2122)**

First of all, we would like to thank the editor and reviewers for their valuable comments. We have taken all the suggested changes into consideration and revised the manuscript accordingly. The reviewers' comments are copied here as texts in BLACK, our responses are followed in BLUE, and the major corrections are marked in RED in the manuscript.

Brown carbon (BrC) is an important component of aerosols in the atmosphere, and there are still significant uncertainties on their chemistry and physical properties as well as their influences on the atmosphere. This manuscript presents an observationally-constrained approach to estimate the radiative effects of BrC aerosols using routine ground-based measurements, and offers a convenient method to assess the climate impacts of BrC. This study effectively integrates observations, optical computations, and radiative transfer models. By employing optical closure techniques, the radiative effects of black carbon were isolated. This approach proves to be both straightforward and efficacious. Meanwhile, by considering only the conventional observations and numerical models, the framework of the proposed method shows great potential for further studies. Overall, the study is well motivated and adds to our understanding of BrC effects. There are several areas that need clarification or revision prior to publication.

**Response**: Thanks so much for your constructive comments. We have implemented all suggestions for improvement in the revised manuscript. Please find our point-by-point responses listed below.

1. More details could be provided on the observations used to constrain the analysis. ln particular, the authors should specify details such as the sampling time period and measurement frequency for each instrument.

**Response:** Thanks. We have added much more descriptions on the observations, such as sampling time period, measurement frequency, and so on (Lines 88 in the revised manuscript).

2. From the perspective of content relevance, it appears more appropriate to position Figure 1 and its associated description with in the Section 2 rather than the third section.

**Response:** Thanks for the suggestion, we have reorganized the manuscript by moving Figure 1 and its associated description to Section 2.

3. Figure 2 serves as a comprehensive overview of the proposed methodology, playing a pivotal role in the exposition of this paper. To provide a clearer understanding, additional space to elucidate the details within Figure 2 is suggested, including the calculation methods for parameters such as MAC (Mass Absorption Coefficient). Alternatively, to manage space constraints, specific algorithms for each subsection of Figure 2 can be referenced in the subsequent sections of this paper.

**Response:** Thanks for the suggestion. We added detailed descriptions of the flowchart in the figure caption, in which the important components in the figure are referred to the subsections of the paper. This will ensure the readers have a clearer understanding of our methodology and calculations.

[Figure]

**Figure 2. Flowchart of estimation of BrC radiative effect. The part with purple background corresponds to the direct observations used, which are detailed in Sections 2 and 3.1. The optical closure part, which uses the direct observations to separate the properties of each type of aerosol (i.e., AOD, SSA, ASY, surrounded by the yellow dotted line), is illustrated by the part with blue background. The bottom yellow part indicates the output for radiative estimations. We adopted a three-component aerosol model (BrC, BC, and pure light-scattering components, i.e., LSC). More details are available in Section. 3.2. After clarifying the properties of each type of aerosol (i.e., AOD, SSA, ASY), the LibRadTran Model is used to estimate the BC and BrC radiative effects.**

4. Figure 4 indicates that the imaginary part of BrC refractive indices may differ over two orders of magnitudes. Would such variation introduce additional uncertainties on the results of this study?

**Response:** Yes, there are significant uncertainties on the refractive indices of BrC. As suggested by both reviewers, we added a sensitive study on the influences of BrC refractive indices (imaginary part) on our BC and BrC radiative effect estimation. We found that BrC refractive index variations may introduce uncertainties up to over 50% on the BrC TOA RF, and more details were added in Section 5.

Page 16 line 364-366 in the main text,
"The right panel shows the influences deriving from uncertainties of the absorptivity of BrC. Except the RF (TOA) with a relatively larger difference of 62%, the rest of the BrC radiative effects (average absolute values) were all below 30%, and, as expected, all the differences were close to zero."

5. In Figure 5, the label "LSC/10" is confusion.

**Response:** To avoid confusion, we redesigned the figure, and the new figure in the form of a double Y-axis becomes much clearer.

6. More discussions on regarding the applicability of the method and the generalizability of the results are suggested, and the limitations of the method could also be discussed.

**Response**: In the article, we use observational data from the Nanjing site as an example to verify the feasibility of the method, but the method can be applied to other regions. Additionally, we acknowledge that the current method still entails a certain degree of uncertainty. We added a new section (Section 5) to discuss and analyze the uncertainties of our method, including the imaginary part of the BrC refractive index, BC particle geometries, and AAE. Furthermore, some discussions on the further works were added in Section 6.

7. It is advisable to further improve the figures qualities. For instance, the tick labels in Figure 4 appear relatively small and could benefit from a consistent font size.

**Response:** Thanks for the suggestions. We have improved all figures and kept a relatively consistent format for them in the revision.

**Some minor comments:**

8. Line 17, To enhance clarity, you can split the sentence into two as follows: "To constrain the total and other aerosol contents, we conducted an optical closure study. Subsequently, the optical properties and concentrations were estimated."

**Response:** Thanks for the suggestion, we have corrected it in the revised manuscript (Line 17 – Line 18).

9. Line 46, "currently, materials such as humic-like substances, polycyclic aromatic hydrocarbons, and lignin are all considered BrC" should be "Currently, materials such as humic-like substances, polycyclic aromatic hydrocarbons, and lignin are all considered as BrC".

**Response:** We have corrected it in the revised manuscript (Line 47).

10. Line 106,"LT" should be "local time".

**Response:** Thanks, and it is corrected (Line 108).

11. The label (a) and (b) in figure 5 is missed.

**Response:** Sorry for the mistake and we have added it to the revised manuscript.

12. Line 337, "that of BC" should be "that caused by BC".

**Response:** We have corrected it in the revised manuscript (Line 383).

---

## Author Comment (AC2)

**Response to Reviewer #2 (EGUSPHERE-2023-2122)**

First of all, we would like to thank the editor and reviewers for their valuable comments. We have taken all the suggested changes into consideration and revised the manuscript accordingly. The reviewers' comments are copied here as texts in BLACK, our responses are followed in BLUE, and the major corrections are marked in RED in the manuscript.

The manuscript employs a hybrid method to estimate brown carbon (BrC) radiative effects using a combination of aethalometer measurements of aerosol absorption, an optical separation method, simulated BrC optical properties, and a radiative transfer model. As BrC is still poorly characterized in the field and largely ignored by many chemical transport models and climate models, the presented results contribute to our understanding of its radiation and climate significance. The manuscript fits the scope of ACP very well. I have some comments below for the authors to address.

**Response**: Thanks so much for your constructive comments. We have implemented all suggestions for improvement in the revised manuscript. Please find our point-by-point responses listed below.

Line 143-145: the authors set the AAEBC=1 to calculate the absorption contributions of BrC at different wavelengths. However, existing research suggests that the AAE of BC in the atmosphere varies within a certain range due to factors such as mixing state and morphology (Lack and Cappa, 2010; Zhang et al., 2020b). To comprehensively assess BrC's absorption contributions, different AAEBC values could be set, and some sensitivity analyses are needed to evaluate the impact of this parameter on the study results.

Lack, D. A. and Cappa, C. D.: Impact of brown and clear carbon on light absorption enhancement, single scatter albedo and absorption wavelength dependence of black carbon, Atmos. Chem. Phys., 10, 4207–4220, https://doi.org/10.5194/acp-10-4207-2010, 2010.

Zhang, X., Mao, M., Yin, Y., and Tang, S.: The absorption Ångstrom exponent of black carbon with brown coatings: effects of aerosol microphysics and parameterization, Atmos. Chem. Phys., 20, 9701–9711, https://doi.org/10.5194/acp-20-9701-2020, 2020.

**Response**: Per your suggestion, we added a new section (Section 5) to discuss the influences of $AAE_{BC}$ uncertainties on our radiative effect estimation. We followed the $AAE_{BC}$ values suggested by Liu et al. (2018), and found that a change of BC AAE from 1.0 to 0.8 may introduce BC radiative effects of approximately 10% and BrC effects of over 40% (due to an obvious increase in BrC amount estimation).

Line 355 – Line 359 in the main text,

"The BC AAE assumption showed influences on BC radiative effects with relative differences of approximately 10%, but much stronger influences on BrC radiative effects, i.e., ~40%. This result was ascribed to smaller BC AEE causing weaker BC absorption in shorter wavelengths, leading to smaller radiative effects. However, larger BrC absorption coefficients and amounts during the segregation could cause larger BrC radiative effects."

Line 178-179: In the optical closure calculation, a three-component aerosol model (BrC, BC, and pure-scattering components) was chosen with the assumption of an external mixing state. However, internal mixing between chemical components has been confirmed as a crucial factor influencing aerosol optical properties. It is recommended that the authors consider the internal mixing state to comprehensively evaluate the aerosol optical properties. Alternatively, the authors could explicitly state the reasons for choosing the external mixing model. This discussion will enhance the reliability of its results.

**Response**: This is an excellent question and suggestion. The mixing structure plays an important role in aerosol optical and radiative property estimation, which has been extensively studied as well. Considering the high complexity of aerosol mixing states, the general observations considered in this study can hardly provide any meaningful information on our model, so we considered this external mixing assumption for the sake of developing a more practical and general method with existing observations. However, we highly agree with the suggestion, and will further investigate the influences of internal mixing on this work in future work. The discussions above have also been added in the revision.

Line 209-210: In this study, it is crucial to address whether the Mie numerical simulation is based on the spherical assumption, as this may introduce biases in optical calculations for non-spherical particles. Particularly, a spherical Mie model tends to significantly overestimate the light absorption of fractal BC particles. The numerical simulation of aerosol optical properties is intricately linked to the accuracy of mass distribution among different components. It is recommended that the potential biases introduced by the spherical assumption should be discussed.

**Response**: As mentioned above, we added a systematic sensitive study on the influences of our assumptions on BC/BrC radiative effect estimations, which includes the influences due to BC geometries. To achieve this, we used the BC optical property database with fractal aggregate structures we developed (Liu et al., 2019), and discussed the differences in the radiative effects between spherical and fractal-aggregate-based

BC particles. The corresponding results and discussions are presented in Section 5 of the revision.

Line 359 – Line 364 in the main text,
"The assumption regarding BC spherical particles showed minor influences on BrC radiative effects, with the relative differences being less than 5%, but much stronger influences on BC radiative effects, particularly on the RF(TOA) with an average deviation of almost 50%, caused mainly by weaker scattering owing to BC non-sphericity (Li et al., 2016). In other words, the assumption regarding spherical BC led to overestimation of BC scattering, increasing the upward radiation reaching the TOA. However, the influence of BC non-sphericity on BrC/BC segregation was lower and the BrC radiative effects change was less than 5%."

As shown in Figure 4, the uncertainty in the Refractive Index of BrC is notable. The authors should provide additional details explaining the reason for choosing the RI reported by Shamjad et al. (2016). It would be beneficial to illustrate the factors such as the similarity in organic aerosol composition and sources between the two cities. This additional information will enhance the understanding of the selection criteria for the RI values.

**Response:** Yes, there are significant uncertainties on the refractive indices of BrC. As suggested by both reviewers, we added a sensitive study on the influences of BrC refractive indices (imaginary part) on our BC and BrC radiative effect estimation. We found that BrC refractive index variations may introduce uncertainties up to over 50% on the BrC TOA RF, and more details were added in Section 5.

Line 364 – Line 367 in the main text,
"The right panel shows the influences deriving from uncertainties of the absorptivity of BrC. Except for the RF (TOA) with a relatively larger difference of 62%, the rest of the BrC radiative effects (average absolute values) were all below 30%, and, as expected, all the differences were close to zero."

Line 205-207: The data utilized in the study are relatively dated, and it is recommended to consider relevant data from recent studies. The inclusion of more recent data would enhance the timeliness and relevance of the findings.

**Response**: Thanks for the suggestion. As you may notice, even this "general" model requires a large number of observations for aerosol constraints. Currently, the relatively dated results are used due to their well-maintained quality and longer observational period. Meanwhile, this study mainly focuses on introducing the method for BrC

radiative effects, and the date of the observations will not influence the application of the methods. Of course, we will further improve and validate our method for more recent data in future studies.

Line 207: The density of BC is commonly reported as 1.8 g·cm$^{-3}$ in the literature. It is advised to verify the accuracy of the statement indicating a density of 1.0 g·cm$^{-3}$ for BC in this context.

**Response**: Thanks for suggestion. After carefully review, we agree that 1.8 g/cm$^3$ may be a more appropriate value, and have updated our results by assuming a BC density of 1.8 g/cm$^3$. The BC density mostly influences the estimated BC mass concentration/amount, while having less effect on our optical closure study.

Line 216 in the main text,

"The densities of BC and scattering aerosol were assumed at 1.8 and 1.7 g·cm$^{-3}$, respectively."

The calculation of radiative forcing (RF) in this study is unclear regarding whether it considers only direct radiative forcing or also includes indirect radiative forcing. Some clarification is needed.

**Response**: Thanks for the suggestion. We have clarified the RF considers only direct radiative forcing in this study.

Line 78 in the main text,

"Radiative forcing (RF, only direct radiative forcing) and influences on the photosynthetically active radiation (PAR) and actinic flux (AF) were considered and discussed."

The English language needs to be further polished. Some necessary edits in the abstract:
- L15, convenience -> efficient, concise -> available
- L23, BrC induces a warming effect with an average instantaneous radiative forcing (RF) of 6.4 ± 3.4 W m-2, corresponding to 29.2% of the BC RF.
  that of black carbon (BC).
- L26, you may want to say "PAR attenuation".

**Response**: Sorry for the language problems, we have corrected the mistakes, and all authors carefully proofread the revision. Meanwhile, the revision was further edited for proper English language, grammar, punctuation, spelling, and overall style by a qualified native English-speaking editor.

Line 15 in the main text,

"This study proposes an efficient method for estimating BrC radiative effects based on the available observational data."

Line 23 in the main text,
"In the atmosphere, BrC plays a warming role, with its average instantaneous radiative forcing (RF) and standard deviation of $4.0 \pm 2.3$W m$^{-2}$ corresponding to $15 \pm 4.2\%$ of the black carbon (BC) RF."

Line 26 in the main text,
"The photosynthetically active radiation (PAR) attenuated by BrC was approximately $33.5 \pm 9.4\%$ of that attenuated by BC."